# Theoretical and Practical Evaluation of the Feasibility of Zinc Evaporation from the Bottom Zinc Dross as a Valuable Secondary Material

**DOI:** 10.3390/ma15248843

**Published:** 2022-12-11

**Authors:** Pauerová Katarína, Trpčevská Jarmila, Briančin Jaroslav, Plešingerová Beatrice

**Affiliations:** 1Faculty of Materials, Metallurgy and Recycling, Institute of Recycling Technologies, Technical University of Kosice, Letna 9, 042 00 Kosice, Slovakia; 2Institute of Geotechnics SAS, Slovak Academy of Sciences, Watsonova 45, 040 01 Kosice, Slovakia; 3Faculty of Materials, Metallurgy and Recycling, Institute of Metallurgy, Technical University of Kosice, Letna 9, 042 00 Kosice, Slovakia

**Keywords:** bottom zinc dross, thermodynamic study, zinc evaporation, inert atmosphere

## Abstract

This study presents a theoretical and practical evaluation of zinc evaporation from bottom zinc dross (hard zinc) as a secondary zinc source (zinc content approximately 94–97%), which originates in the batch hot-dip galvanizing process. The thermodynamics of the zinc evaporation process were studied under the normal pressure (100 kPa) in the inert atmosphere, using argon with flow rate 90 mL/min. Samples were subjected to the evaporation process for 5, 10 and 20 min under the temperature of 700 °C and 800 °C, respectively. For the theoretical thermodynamic study, HSC Chemistry 6.1 software was used and final products, as well as residuals after the evaporation process, were analyzed by SEM (Scanning Electron Microscopy) and EDX (Energy Dispersive X-ray). Calculated and experimental argon consumption in the process of zinc evaporation has been compared. A high purity zinc with efficiency over 99% was reached. Due to a dynamic regime, argon consumption at the temperature of 700 °C and 800 °C were 7 times and 3 times, respectively, less than calculated.

## 1. Introduction

Zinc, as the fourth most produced metal worldwide, is a key element of industrial society. Zinc has a wide range of usage in crucial applications such as corrosion protection of steel in buildings, infrastructure, and vehicles [1,2]. Within these applications zinc is used in many forms such as high grade metal (coatings for steel), alloying elements (in brass), and chemical compounds, e.g., zinc oxide in tires [3,4]. In 2021, the world zinc reserves were estimated to be some 250 million tons. Australia owns the largest zinc reserves worldwide—an estimated 69 million tons [5]. Global zinc mine production reached nearly 13.8 million tons in the year 2021. China is the largest zinc miner and metal producer with 30.4% of the global amount [6]. In the same year, approximately 14.05 million tons of refined zinc was consumed worldwide [7].

Zinc, at the end of a product’s life, can be recovered and recycled without the loss of its characteristics or value. Zinc containing products become a source of recycling feedstock at the end of their life, known as “old scrap”. The old scrap is collected and processed based on scrap availability, metal composition and ease of processing. During the processing phase, zinc becomes available for recycling, as a “new scrap”, due to potential losses during manufacturing and fabrication, e.g., drosses, residues, off-cuts. The current zinc end of life recycling rate reaches 34% [8,9,10].

Presently, approximately 70% of the zinc produced originates from mined ore and 30% from recycled or secondary zinc. More than 50% of zinc is used in galvanizing industry as zinc coatings. The hot-dip galvanizing process (HDG) is the most used method of zinc coating applications [10,11,12].

The hot-dip galvanizing process generates several wastes or secondary products, respectively. Solid wastes are more valuable due to its high zinc content. One kind of such waste is bottom zinc dross or so-called “hard zinc”. Bottom zinc dross is created in the galvanizing process as the maximum iron solubility is exceeded, which is 0.03% in the molten zinc under the process temperature (approximately 450 °C). After exceeding this solubility, iron starts to precipitate into intermetallic compound in the form of FeZn_13_. As intermetallic compound has a higher density than molten zinc, these compounds settle down on the bottom of the zinc kettle. Intermetallic compounds, together with molten zinc, are regularly withdrawn from the zinc kettle by a special mechanical dipper bucket. Bottom zinc dross is considered to be a valuable secondary zinc source due to its high zinc content (94–97%) [13,14].

Zinc recovery techniques are mainly based on the character of the zinc waste (metallic form, oxidic form, or complex compound). Pyrometallurgy, hydrometallurgy or its combination are used. Hydrometallurgy is preferred when zinc content is lower, due to economic advantages. Hydrometallurgical treatment, generally, involves the leaching—solvent, extraction—electrowinning route for the zinc recovery in the metallic form, as well as in the form of zinc compounds [15,16,17,18,19]. Several hydrometallurgical methods were devised to reclaim zinc values from zinc waste, generally. Hesham and Kamaleldin [20] focused on the extraction of zinc from blast-furnace dust using ammonium sulfate. Zinc oxide content in the studied sample was 49.6%. Zinc was presented in the form of hydrozincite, hemimorphite, smithsonite and sphalerite. In the first place, the dust sample was roasted at 850 °C to obtain a more stable zinc form (ZnO) that is more susceptible towards ammonium sulfate leaching at low temperatures. Under the set optimal conditions for the formation of soluble zinc compounds, with molar ratio 1:8 of roasted zinc dust (ZnO) and (NH_4_)_2_SO_4_ under the temperature of 350 °C, up to 95% of zinc as hydroxide was leached with 0.5 M sulfuric acid. Wang et al. [21] leached basic oxygen steelmaking filter cake in organic acids (oxalic, citric, acetic, propionic, butyric and valeric) to gain zinc from sample, selectively. The content of zinc in the filter cake was on average 6.5% in the form of zinc oxide and zinc ferrite. Butyric acid exhibited excellent selectivity among tested acids, with up to 49.7% of the zinc being leached from the filter cake. Mehmet et al. [22] studied optimization possibilities of selective zinc leaching from electric arc furnace steelmaking dust using response surface methodology. Zinc content in the sample was 26.95% in the form of ZnO and complex oxidic compounds bonded with iron, manganese, magnesium, or titan. The most influential zinc leaching factors were determined as acid concentration and quadratic factors of acid concentration by using Anova. The proposed criteria in which zinc recovery is greater than 70% and iron recovery is lower than 10% was: acid concentration between around 1.6–3.1 mol·L^−1^, leaching duration of 56.42 min and L/S ratio of 5. Xie et al. [23] studied zinc extraction from industrial waste residue by conventional acid leaching. Zinc content in the waste was 24.27%. The effect of reaction time, sulfuric acid concentration, leaching temperature, stirring speed, and liquid solid ratio on zinc leaching rate were studied. Authors reached 86.34% of zinc recovery under the condition of sulfuric acid concentration 0.61 M, reaction temperature 25 °C, liquid-solid ratio 4:1 (mL/g), stirring speed 400 rpm, and leaching time 30 min. Radzyminska-Lenarcik et al. [24] recovered zinc from metallurgic waste sludges. Zinc bearing sludge contained 11.0–13.0% of zinc. Hydrochloric, sulfuric, and lactic acids, as well as ammonia and NaOH solutions were studied for the highest amount of zinc recovery. Extraction methods such as electrolysis and solvent extraction were applied. The most effective leaching solutions were the concentrated ammonia, 30% NaOH, and 80% lactic acid. By electrolysis 92–99% of metal zinc was recovered, and using solvent extraction recovered 96–99% of zinc depending on the solution pH.

Lorenzo et al. [25] recovered electrolytic zinc or zinc sulfate from galvanizing zinc dross by means of acid leaching, followed by using solvent extraction. The author declared patent for this technique, and its research is extensively described in detail. Galvanizing dross with zinc content of 63%, of which 3% is metallic zinc, was put under the leaching process in organic acid reagent of D2EHPA under the temperature of 40 °C. The zinc reaching yield was within 50 min and pH 2.7 was 97.7%. Besar et al. [26] processed zinc dross originating in the hot-dip galvanizing by hydrometallurgy to obtain zinc oxide. Zinc content of the dross was 96.5% in its metallic form. Variables as process time (30, 60 and 90 min), temperature (150 °C, 170 °C, 190 °C), concentration of glacial acetic acid (20%, 40%, 60%) were studied. The best process conditions were at 90 min under the temperature of 150 °C, and a glacial acetic acid concentration of 60%. Under set conditions, a yield of 62.45% was obtained and zinc content in zinc oxide was 70.23%. However, the pyrometallurgy seems to be more reasonable when zinc concentration is higher [16,17,18,19]. The best available pyrometallurgical technology for zinc-bearing residue treatment is the Waelz process using a rotary kiln. The Waelz process is characterized by the volatilization of zinc from an oxidized solid mixture (zinc bearing waste) through reduction by coke or coal. The Waelz kiln operates at a rotational speed of approximately 1 rpm and a slope of 2–3%. The air enters the kiln at the slag discharge end. Solid charge is dried and heated up until a reaction starts. At about 800 °C zinc oxides start to reduce and thus zinc volatilizes into a gas phase. Subsequently, zinc vapors are re-oxidized into ZnO and transported as solid dust with the counter current process air flow into a dust chamber [27]. The Waelz kiln process was also used for the carbothermic reduction of zinc containing waste by author Zhang et al. [28]. In the research zinc oxides from electric arc furnace dust were investigated. Zinc content in the dust was 21.5% in the form of franklinite and zincite. Dust was reduced by carbon at temperatures between 800–1300 °C. The ZnO reduction and zinc evaporation occurred in the temperature range of 1000–1100 °C. At a temperature of 1100 °C, 99.11% of zinc was evaporated. Xue Denga et al. [29] applied the evaporation method on the blast furnace dust, followed by condensation and separation with vacuum carbothermal reduction into the metallic zinc. The content of ZnO in the dust material was in the amount of 7.9%, of which metallic zinc was 6.34%. Authors conducted the first group of vacuum carbothermal reduction at the temperature of 800 °C, within the time of 90 min and carbon additions of 8%, 10%, and 12%. The second group of experiments was conducted with a reduction temperature of 900 °C, within the time of 90 min and carbon additions of 6%, 8%, and 8% with 2% B_2_O_3_ additive. When the range of the carbon addition was 10–14% and the temperature reached 900 °C, the volatilization rate of metallic zinc exceeded 99.6%. Jintao et al. [30] processed copper smelter dust containing 73.89% of ZnO by evaporation and subsequent condensation of zinc vapors from dust. According to their results, 99.94% of zinc oxide powder was transformed into zinc vapors through carbothermal reduction at the temperature of 1373–1573 K for 30–60 min. Zinc recovery from a manganese battery was also investigated by Zhan et al. [31]. Manganese batteries with zinc content of 14.32% were processed by vacuum evaporation under the temperature of 1123 K and oxygen-control oxidation with 12.5% oxygen content and 21 L/min nitrogen flow rate to prepare a nano-zinc oxide with high added value. Zinc dross from the hot-dip galvanizing process was studied by author Wang et al. [32] through the method of super-gravity separation. Zinc dross contained 65.4% of zinc, mostly presented as pure metallic zinc, and the rest zinc was bonded in the intermetallic compounds form of Fe_2_Al_5_Zn_0.75_. Studied factors were gravity coefficient (15, 50, 100, 300, 500, 800, 1000), separating time (15, 60, 120, 180, and 350 s), and separating temperature (430, 460, 510, 560, and 610 °C). Over 79% of zinc was recovered with a high purity of about 99% at gravity coefficient 500, within a time of 180 s and temperature of 510 °C.

Bottom zinc dross from the hot-dip galvanizing process was put under investigation in several studies. Bottom zinc dross was investigated using electrochemical and pyrometallurgical methods by Ghayad et al. [33]. Dross containing 96.18% of zinc was dissolved in concentrated sulfuric acid to create zinc sulfate solution (100 g Zn/L). Dissolution was performed under the temperature of 55 °C for 1.5 h. Solution was adjusted by hydrogen peroxide to remove iron. The obtained filtrated zinc sulfate solution (with content of Zn^+2^ in amount of 32.1% besides Al^+3^ and Fe^+2^ in amount of 0.99% and 0.53%, respectively) was then used for electrowinning and electrorefining of zinc. Authors studied variable factors such as current density (40–90 mA/cm^2^ rising by 10 mA/cm^2^) and time (120, 270, 360, 480, and 600 min). The highest zinc efficiency of 90% was obtained at the current density of 40 mA/cm^2^, within time of 30 min and under the temperature of 25 °C. Zinc deposit was in high purity of 99.99%. A second research technique of zinc recovery (same author’s research [34]) using pyrometallurgical technique was conducted in the heating furnace. The dross sample was placed in the heating crucible, covered with thermal cement, and heated to the temperature of 900 °C. Under the given temperature formed zinc vapors passed through the silicon carbide tube and subjected to cooling into zinc melt. The studied parameter was a temperature and time of evaporation process. The highest zinc recovery (82%) was obtained under the temperature up to 1000 °C within 1.5 h. Further increases in time were not effective in higher zinc recovery. Obtained zinc was of high purity (99.95%). Prasad [34] prepared electrolytic zinc powder from bottom zinc dross in the sodium zincate solution. Content of zinc in the studied sample was 94.5%, presented in the form of metallic zinc and the intermetallic compound of Fe-Zn. To obtain best results, the author studied several parameters influencing electrowinning process, such as effect of zinc concentration in the electrolyte (16, 30, 45, 60, and 75 g/L of zinc), effect of NaOH concentration (180–220 g/L of NaOH), effect of anode to cathode distance (20–60 mm), temperature within time (30–50 °C within 0–125 min), effect of current density on the electro deposition of zinc powder (2–10 amp/dm^2^), and the specific power consumption. A high purity of zinc powder (99.5%) was obtained at room temperature in efficiency of 86% at 5 amp/dm^2^ current density, 1.2 V impressed voltage, concentration of electrolyte 16 g/L Zn and 220 g/L NaOH. Sinha et al. [35] established a leaching-precipitation-crystallization route to produce high-grade zinc sulfate and phosphate salts, along with by-products. Zinc content in the studied sample was 97.5%. Firstly, zinc dross was leached in a sulfuric acid at a specified concentration (9–12% *v*/*v* was varied) at ambient temperature for 24 h. After this process, the leach liquor was drained out from the leaching tank and filtered. Then, liquor was put under the precipitation process to remove iron and other impurities. The purified liquor was further used for crystallization of zinc sulfate salts (temperature set at 70 °C and 30 °C for stability range of crystallization of ZnSO_4_·H_2_O and ZnSO_4_·7H_2_O, respectively) and precipitation of zinc phosphate salts (considering pH ~3.5 and ~4.5, flower shape, sheet-like, and spherical aggregate lumps like morphologies were observed).

In practice, the production of ZnO is the most common way of zinc recovery from the bottom zinc dross. There is no established process for bottom zinc dross treatment to zinc recovery in its metallic form to keep the zinc in the galvanizing loop. The present study focuses on the development of the most effective and economical way for high purity zinc recovery from bottom zinc dross by zinc evaporation and its condensation. This can be achieved by detailed research of the zinc evaporation process in the dynamic system using a lower temperature than of the zinc boiling point. Before the overall process can be evaluated in terms of economic viability, several tasks and partial investigation must be conducted. The first main task is a thermodynamic study of the zinc evaporation process from bottom zinc dross. For this purpose, thermodynamics were used as a preliminary study for an effective and economical way of successful zinc recovery from the bottom zinc dross. Theoretical and experimental results were evaluated.

## 2. Experimental Section

### 2.1. Materials

Samples of bottom zinc dross (Figure 1a) were drilled, and zinc chips were obtained for experimental usage (Figure 1b).

The sample was analyzed to determine zinc and the content of other elements (Table 1) through AAS (atomic absorption spectrometry) analysis using model ContrAA 700. Phase analysis was realized by XRD (X-ray diffraction) using model XPERT PRO RV-11/2010, PANalytical, as shown in Figure 2 and Table 2. Microstructure of bottom zinc dross using light microscopy (Olympus BX53M) after etching can be seen in Figure 3.

The light objects are intermetallic compounds of FeZn_13_. The dark color represents a zinc matrix that was formed by the solidification of excess zinc melt, which was captured when removing intermetallic compounds (FeZn_13_) from the bottom of the galvanizing kettle.

### 2.2. Analysis of Zinc Evaporating Conditions

A thermodynamic study available in the literature and by HSC Chemistry software 6.1 was carried out to set experimental conditions. Parameters such as zinc evaporation temperature, inert gas addition, time of evaporation, and zinc condensation were studied.

The theoretical amount of inert gas (e.g., Ar) needed for evaporation of one gram of Zn (0.0153 mol Zn) is calculated from the thermodynamic data (Table 3; HSC 6.1) according to the Equation:(1)$$n_{\mathrm{Ar}}=\frac{n_{\mathrm{Zn}}\cdot p_{\mathrm{Ar}}}{p_{\mathrm{Zn}}},\;\;[\mathrm{mol}]$$
and recalculated to Ar volume at 25 °C (V_Ar_ = n_Ar_·V_m_(25 °C)). n_Ar_—amount of substance [mol], n_Zn_—amount of substance of Zn(g) vapors [mol], p_Ar_—equilibrium vapor pressure of Ar [Pa], V_Ar_—volume of inert Ar gas [dm^3^], V_m_(25 °C)—24.5 dm^3^ by one mol of ideal gas. Calculated parameters data within selected temperatures are listed in Table 3.

The melting temperature of Zn is 419.5 °C. With increasing temperature, the partial zinc vapor pressure over the zinc melt increase. Reaching temperatures 750–800 °C, zinc starts to evaporate in greater amounts. Reaching zinc boiling point (907 °C), the zinc vapor curve rises rapidly. Inert Ar gas prevents oxidation of zinc vapors during the evaporation process.

The equilibrium state of Zn in system (0.0153 mol Zn: 0.001 mol Ar) and (0.0153 mol Zn: 1 mol Ar) in the temperature range of 500–950 °C is compared in Figure 4a,b (HSC 6.1 diagrams). The calculation of p_Zn_ dependence on temperature was conducted regarding how condensed Fe impurities affected the activity of Zn(l) (a ≠ 1).

Decreasing the overall pressure in the system (below 100 kPa) moves the zinc boiling point to lower values. Similarly, it also works with the inert gas flow over the molten zinc surface. In the reactor, the passing inert gas takes zinc vapors out of the evaporation zone, and thus constantly disrupts the effort to achieve equilibrium partial pressure in the system. It means that the addition of a greater amount of argon into the system allows evaporation of a greater amount of zinc. The larger the difference between the equilibrium and non-equilibrium state, the faster the Zn(g) evaporates.

### 2.3. Experimental Procedure

For the experimental process of zinc evaporation, the pipe resistance laboratory furnace was used. The apparatus for evaporation and subsequent condensation of zinc vapors is schematized in Figure 5.

A bottom zinc dross sample in the amount of approximately 3 g was put into quartz tube with a diameter of 12 mm. This quartz tube (12 mm) was then inserted into a larger quartz tube with a diameter of 20 mm (according to Figure 5). Tubes were partly inserted into the furnace. The sample was inserted into the furnace when required temperature was reached (700 °C and 800 °C, respectively). During heating, all apparatuses were blown using argon to ensure inert atmosphere. The argon flow was set up on a flow rate of 90–100 mL/min (mostly 91 mL/min—dependent on the process changes). Evaporation process temperatures were 700 °C and 800 °C, respectively. The time of the evaporation process was observed for periods of 5, 10 and 20 min. At first, the initial experiments under the temperature of 700 °C were conducted to observe sample behavior in the furnace during the evaporation process. The thermodynamics of zinc evaporation showed only feasibility of the evaporation, not the overall time of the zinc evaporation process. After the process, samples were cooled down by removing quartz tube from the furnace at a speed of 2 cm/min, keeping the argon atmosphere. Finally, the theoretical and experimental efficiency of zinc evaporation was compared. The final product (zinc), as well as residue, were evaluated by SEM and EDX.

## 3. Results and Discussion

### 3.1. Thermodynamic Study

Based on the thermodynamic calculation of zinc evaporation and the argon consumption within temperatures (Table 3), two temperatures were chosen for experiments (700 °C and 800 °C). In terms of a detailed simulation of the process, the comparison of the dependence of zinc vapors equilibrium pressure on the argon amount in the system was simulated by HSC software for the temperature of 700 °C and 800 °C, respectively.

Figure 6a,b shows the simulation of zinc evaporation in the argon flow atmosphere at temperature 700 °C (Figure 6a) and 800 °C (Figure 6b), respectively. The calculation in HSC software presumes that condensed phases are impure (a ≠ 1). Inert atmosphere ensures there is no zinc oxidation during the process.

From the graph in Figure 6a, when considering impure zinc in the system, it is necessary to supply ≈0.5 mol (≈12.4 L at 25 °C) of argon to evaporate almost one gram of zinc at the temperature of 700 °C. Increase of temperature to 800 °C (Figure 6b) decreases spent Ar amount by four times (≈0.12 mol, which is ≈2.97 L at 25 °C). Comparing the evaporation process by HSC software (Figure 6) with the calculation from Table 3, there is a difference in the spent Ar argon. Calculation presented in Table 3 is based on a pure zinc system, whereas the calculation made by HSC software considers the system to be impure (iron content). Experiments were conducted with a sample of 3 g, so theoretically there should be 18.6 L and 1.2 L of argon consumption within the temperature of 700 °C and 800 °C, respectively (according to the evaporation process simulation as shown in Figure 6). Real argon consumption is listed in Table 4.

Generally, the size of chips determines the rate of zinc evaporation. The smaller the particles are, the bigger the reaction surface is, and thus the higher the rate of evaporation. Impurities in the chips may also influence the evaporation process. The main impurity is iron bonded with zinc into an intermetallic compound. According to Marder ([37] p. 10–12) during the heating, the zeta (ζ) phase, FeZn_13_, decomposes within the temperature of 530 °C creating delta (δ) phase, FeZn_10_. Then delta phase decomposes into gamma (γ) phase, Fe_3_Zn_10_, within the temperature of 665 °C. Gamma phase is stable within the temperature of 782 °C. Exceeding this temperature, phase decomposes into liquid Zn and αFe. Based on the knowledge of individual phase decomposition, it can be deduced that evaporation process over the temperature of 700 °C is partially influenced by the presence of gamma phase. Zinc bonded into an intermetallic compound evaporates by smaller velocity than pure zinc does. Presence of impurity (iron) slows down the overall zinc evaporation process. The evaporation process under the temperature of 800 °C should not be influenced by the intermetallic compound’s presence, as the zinc is released from the intermetallic compound with iron. However, generally, during the zinc evaporation, iron concentrates presented in the residue. As this iron concentrates in the residue and at the same time the zinc reduces (in the input dose), the final phase of zinc evaporation slows down with time.

### 3.2. Experimental Part

The condensation area of zinc product as well as residue after evaporation (in the quartz tube) are shown in Figure 7. During the process, two forms of zinc product were formed (Figure 8). Zinc foil as well as zinc drops were formed when zinc vapors condensed. The reason for different zinc products in the single evaporation process is probably due to unstable argon flow rate during the evaporation process. When argon flow rate increased, zinc foil formed.

Morphology of analyzed samples can be seen in Figure 9a,b (zinc foil), Figure 10a,b (zinc drops), and Figure 11a,b (residue after evaporation). The description of morphology is below.

Results of SEM analysis of the product in the form of “foil” is shown in Figure 9a,b. The morphology of this product is smooth and consistent.

Results of SEM analysis of the product in the form of “drop” is shown in Figure 10a,b. Spherical products of various sizes have a smooth surface.

Results of SEM analysis of the residue is shown in Figure 11a,b. In Figure 11a, the “needle-like” morphology of residue after evaporation can be observed.

EDX analysis of the zinc foil, zinc drop, as well as the residue after evaporation process under the temperature of 700 °C are shown in Figure 12 ((a) zinc foil, (b) zinc drop, (c) residue).

Zinc products (Figure 12a,b) are high grade purity without any oxidation. No other elements were evaporated during the process. All iron and other elements remained in the residue (Figure 12c). Zinc content in the residue was negligible.

The morphology of the zinc product sample, as well as the residue after the evaporation process at the temperature of 800 °C, can be seen in the Figure 13 ((a) zinc drop, (b) residue).

The result of SEM analysis of the zinc product is shown in Figure 13a, where the individual and connected spherical particles can be observed. The result of EDX analysis of these samples is shown in Figure 14 ((a) zinc drop, (b) residue).

As it is shown in Figure 14a, a high purity zinc product was obtained. No impurities passed to the final product. A negligible amount of zinc remained in the residue (Figure 14b).

Experimental results within selected temperatures and time of evaporation can be seen in Table 4.

From Table 4, a high zinc efficiency was obtained under the set evaporation times. Within the temperature of 700 °C, the time of 10 min was sufficient for evaporation of 92% zinc. This level of efficiency is about to be considered to energy consumption in half the time of evaporation process. During the process of zinc evaporation, iron concentrates in the residue and thus the zinc evaporation may slow down. Comparing zinc efficiency within the temperature of 700 °C and 800 °C in Table 4, it cannot be clearly stated that evaporation at the temperature of 700 °C is more efficient (100% of Zn) than the process under 800 °C (99% of Zn) within the same evaporation time (20 min). Differences in the zinc efficiency are very small, almost negligible, considering the input dose amount, and possible heterogeneity of the sample in the realized experiments. Within the temperature of 800 °C, all zinc was evaporated within the time of 20 and 10 min. Argon consumption for each experiment is also stated in Table 4. The real argon consumption (1.8 L of Ar_(g)_ by 700 °C and 0.9 L of Ar_(g)_ by 800 °C) within the dynamic experimental configuration and the argon gas flow of 90 mL/min trough 12 mm quartz tube is significantly lower than theoretical argon consumption (12.4 L of Ar_(g)_ by 700 °C and 2.97 L of Ar_(g)_ by 800 °C) calculated for equilibrium state in the static test configuration. This significantly lower carrier argon consumption is a result of dynamic regime of the experiment, and thus relates to the velocity of zinc vapors removal from the melt zinc surface in the evaporation zone. The gas flow permanently disrupts the equilibrium state over the molten zinc surface and accelerates the rate of its evaporation significantly. The volume and the velocity of the argon gas is an important parameter for the evaporation process itself. The velocity of the gas flow depends on the flow of the gas, as well as the quartz tube diameter.

The results of pilot experiments confirmed that high pure zinc from bottom zinc dross can be obtained by evaporation and its condensation, quite simply. In the flow of argon gas within a temperature of 700–800 °C the process runs quickly. The zinc evaporation rate depends on both the temperature and the velocity of inert gas flow above the melt. With the increasing velocity of the gas flow, the condensation area of vapors will also shift further along from the evaporation zone. The molten zinc surface is another important parameter influencing the process efficiency.

As expected, during experiments significantly lower amounts of inert gas was consumed, due to the dynamic mode of the process. In this case, it was proved that it is possible to apply decreasing temperature, thus decreasing argon flow rate for process optimalization.

Within this given apparatus configuration (Figure 5), either the flow rate of argon or temperature can be decreased and thus the argon consumption can be optimized. The conducted experiments, selected conditions and test arrangement are not sufficient for a kinetic study of zinc evaporation and the complex evaluation of the overall process efficiency. However, obtained results indicate experimental proceedings for further kinetic study. The conducted research is crucial for kinetics of zinc evaporation as a significant factor for the overall process of optimalization. Kinetic research can address several points, such as argon flow rate within selected quartz tube diameter, defined input surface of the sample, temperature, and time of evaporation process.

In the case of a successful result of the bottom zinc dross refinement and its economic feasibility, this process could be implemented on a semi-operational scale in cooperation with a bottom zinc dross producer. Producers are currently interested in this ongoing research. This research is applied research for a specific type of waste refinement.

## 4. Conclusions

Based on a thermodynamic and experimental study of zinc evaporation from the bottom dross, the following conclusions were derived:The thermodynamic study pointed to the choice of zinc evaporation temperature due to argon consumption (700 °C and 800 °C, respectively);Based on theoretical calculation, the argon consumption at a temperature of 800 °C is four times lower (2.97 L of Ar(g)) than that at a temperature of 700 °C (12.4 L of Ar(g));A high purity zinc product (100% of zinc) can condense in the form of “foil” or “drop” based on the argon flow rate;Argon consumption during the experimental procedure was significantly lower than theoretically calculated due to the dynamic regime. Real argon consumption at a temperature of 800 °C was 0.9 L and at a temperature of 700 °C was 1.8 L, respectively.

Nowadays, the processing of bottom zinc dross in a recycling loop plays an important role for sustainability of the secondary materials. As pure zinc is to be obtained under the economic viable condition, its reuse in the HDG process is ensured.

## Figures and Tables

**Figure 1 materials-15-08843-f001:**
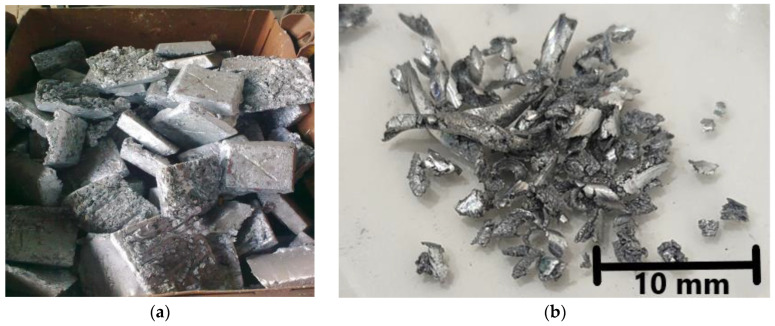
(**a**) Bottom zinc dross sample with dimension of approximately 20 × 20 × 7 cm and (**b**) drilled zinc chips as dose for experiments.

**Figure 2 materials-15-08843-f002:**
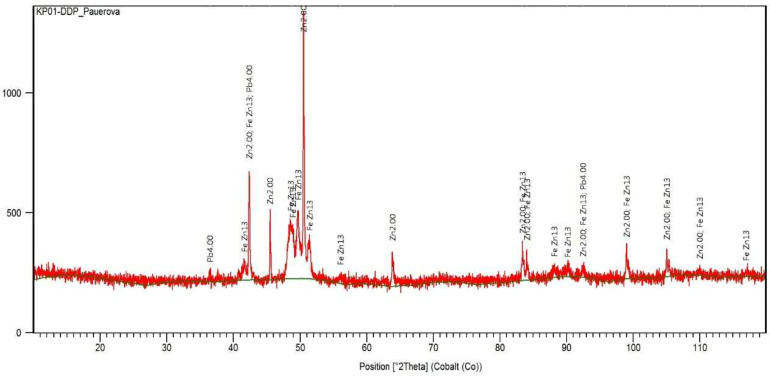
XRD pattern of the bottom zinc dross sample [36].

**Figure 3 materials-15-08843-f003:**
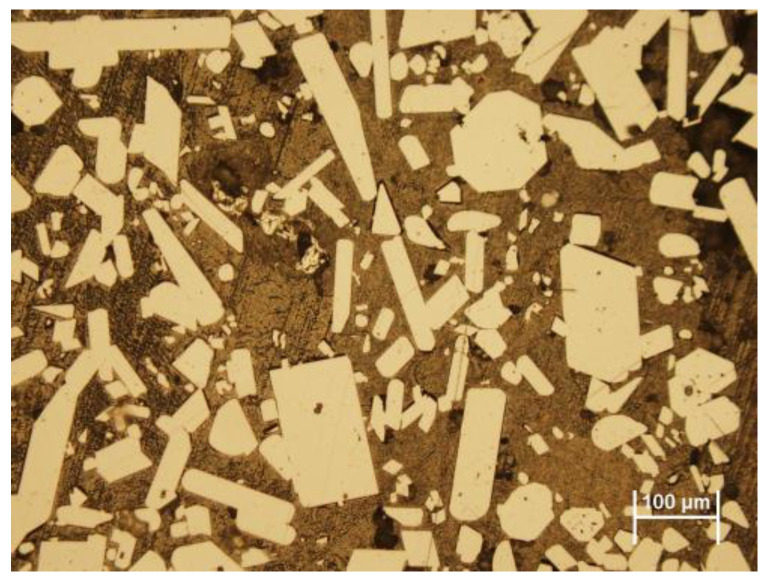
Microstructure of bottom zinc dross after etching (dark color represents a zinc matrix and light color objects represent intermetallic compounds of FeZn_13_).

**Figure 4 materials-15-08843-f004:**
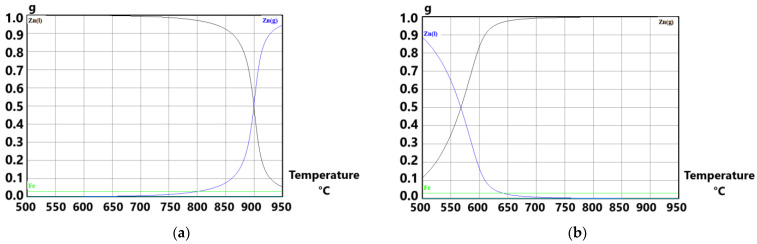
Change of p(Zn) with temperature in the system of (1 g Zn: 0.03 g Fe: Ar): (**a**) 1·10^−3^ mol Ar and (**b**) 1 mol Ar (1 g Zn = 0.0153 mol).

**Figure 5 materials-15-08843-f005:**
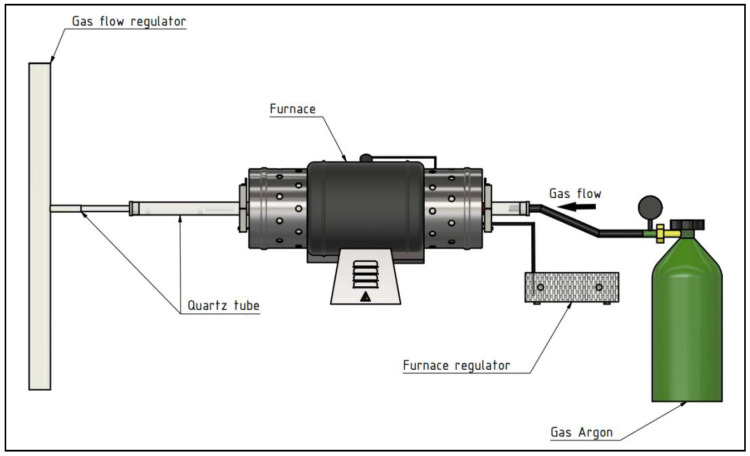
The scheme of apparatus built for experiments: (from left to right) gas flow regulator, quartz tube of diameter 20 mm, quartz tube of diameter 12 mm (inserted in the 20 mm quartz tube), the pipe resistance furnace, furnace regulator, and gas bomb.

**Figure 6 materials-15-08843-f006:**
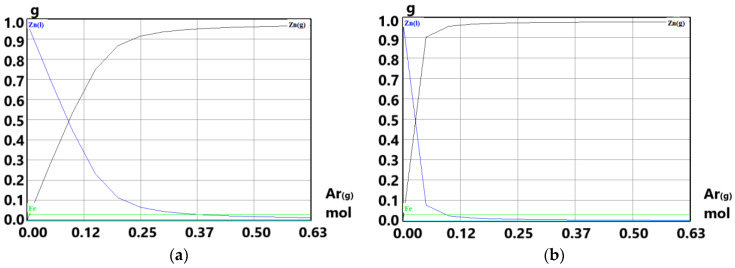
Comparison of dependence of Zn(g) equilibrium partial pressure on the Ar amount in system (1 g Zn: 0.03 g Fe: Ar): (**a**) under the temperature of 700 °C and (**b**) under the temperature of 800 °C. Condensed phases are impure (a ≠ 1).

**Figure 7 materials-15-08843-f007:**
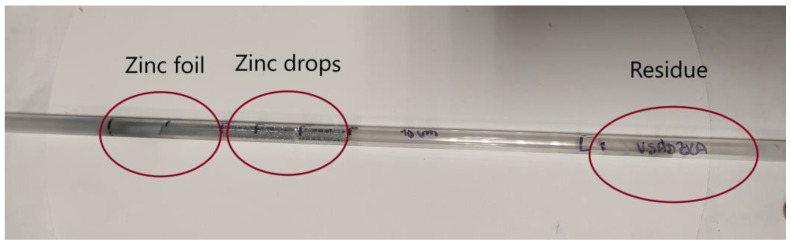
Quartz tube with a dose after evaporation process: zinc foil, zinc drops, and the residue.

**Figure 8 materials-15-08843-f008:**
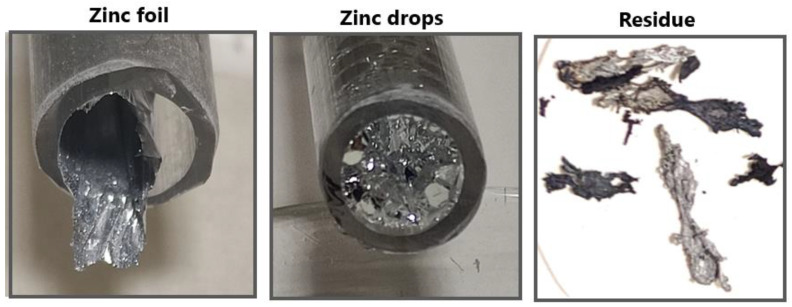
Zinc products (zinc foil, zinc drops) and residue.

**Figure 9 materials-15-08843-f009:**
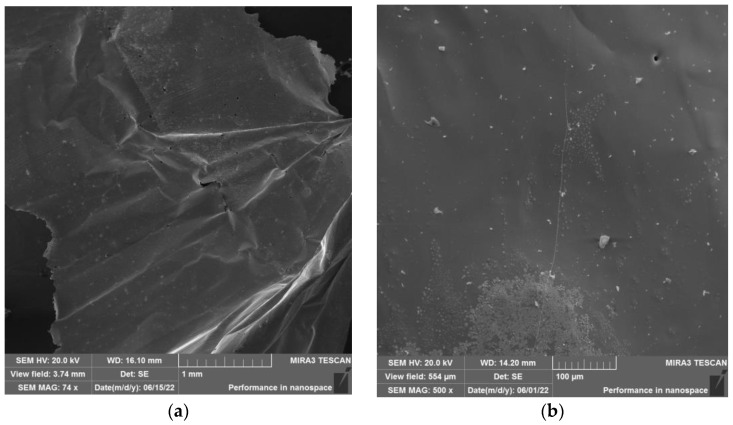
Morphology of the obtained zinc product (evaporation process under the temperature of 700 °C) in form of “foil” at (**a**) 1 mm magnification and (**b**) 100 µm magnification.

**Figure 10 materials-15-08843-f010:**
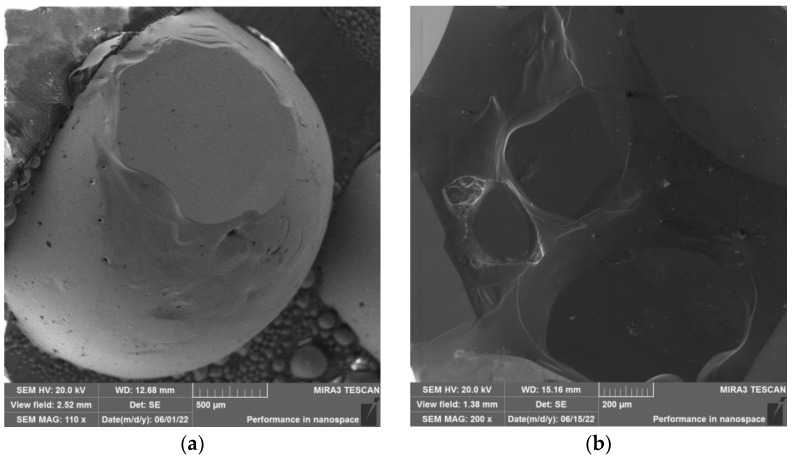
Morphology of the obtained zinc product (evaporation process under the temperature of 700 °C) in the form of “drop” at (**a**) 500 µm magnification and (**b**) 200 µm magnification.

**Figure 11 materials-15-08843-f011:**
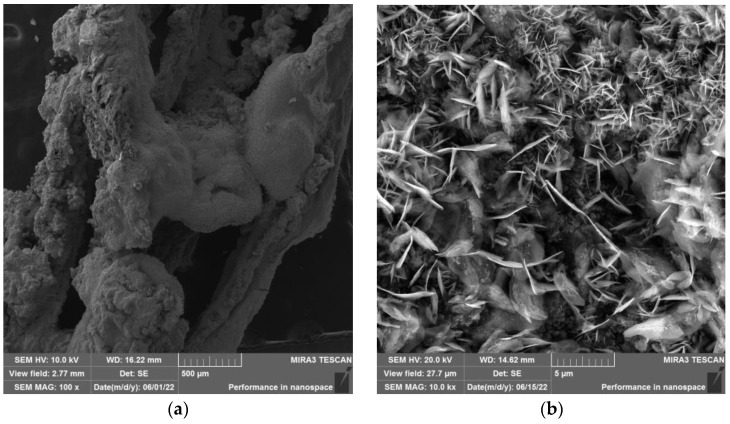
Morphology of the residue after the evaporation process under the temperature of 700 °C at (**a**) 500 µm magnification and (**b**) 5 µm magnification.

**Figure 12 materials-15-08843-f012:**
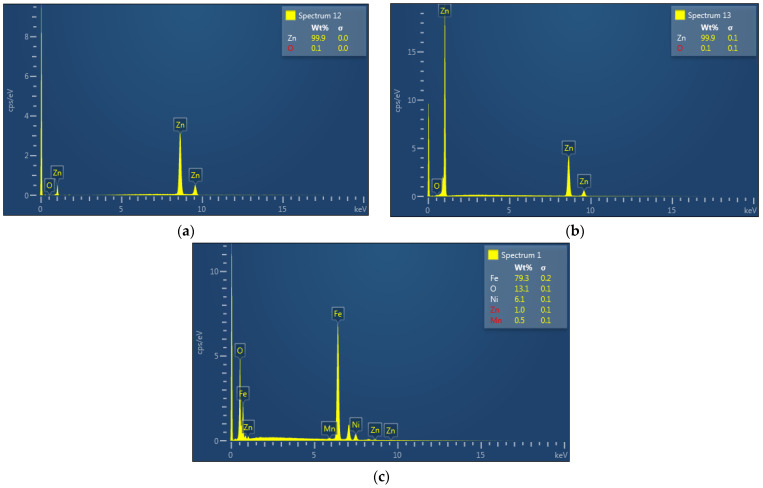
EDX of (**a**) zinc foil and (**b**) zinc drop at 700 °C. (**c**) EDX of the residue after evaporation at 700 °C.

**Figure 13 materials-15-08843-f013:**
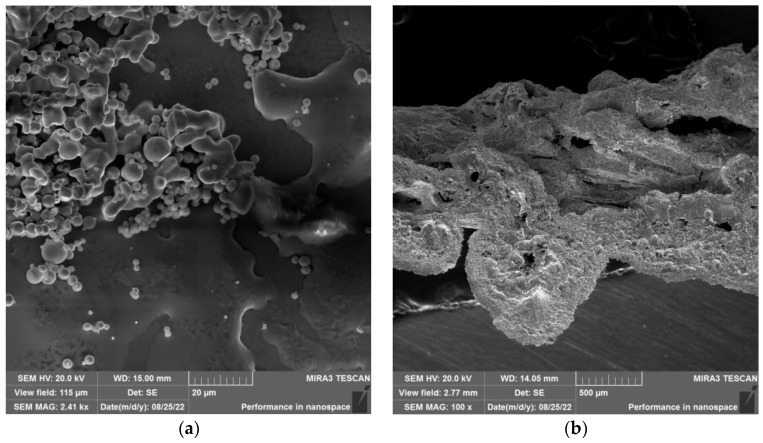
Morphology of (**a**) zinc drop product and (**b**) the residue after evaporation by 800 °C.

**Figure 14 materials-15-08843-f014:**
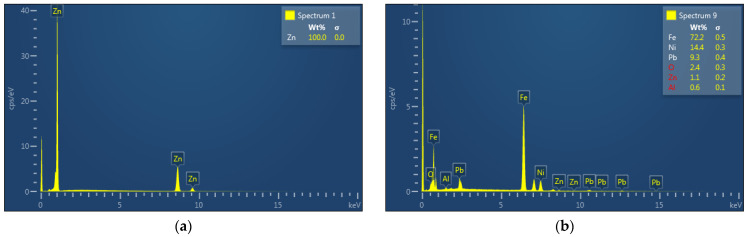
EDX of (**a**) zinc drop product and (**b**) the residue after evaporation by 800 °C.

**Table 1 materials-15-08843-t001:** Chemical composition of the bottom zinc dross sample by AAS analysis.

Element	Zn	Fe	Pb	Al	Ni
Amount [%]	>94–97	2.2–3.4	0.7–1.5	0.8–1	0.3–0.7

**Table 2 materials-15-08843-t002:** Results of the XRD analysis—phase composition of the sample.

Ref. Code	Compound Name	Chemical Formula
96-901-1600	Zinc	Zn
03-065-1238	Iron Zinc	FeZn_13_
96-900-8478	Lead	Pb

**Table 3 materials-15-08843-t003:** Inert gas amount (nAr, V_Ar_) at equilibrium state in system (Zn+Ar) after the evaporation of 1 g Zn (i.e., 0.0153 mol Zn) at selected temperatures.

Temperature [°C]	Equilibrium Constant of Phase Transformation Zn(l) = Zn(g) [-]	Equilibrium Partial Pressure [Pa]		Amounts of Inert Ar Gas [mol]	Volume of Ar at 25 °C [dm^3^]
**t**	**K**	**p_(Zn)_**	**p_(Ar)_**	**n_Ar_**	**V_Ar_ (25 °C)**
**500**	0.00183	183	99,817	8.4	204
**600**	0.0153	1530	98,470	0.98	24
**700**	0.0814	8140	91,860	0.17	4.2
**800**	0.314	31,350	68,650	0.033	0.82
**900**	0.949	94,940	5060	0.0008	0.02

**Table 4 materials-15-08843-t004:** Zinc efficiency within temperature and time of evaporation.

Temperature [°C]	Time [min]	Initial Dose of Sample [g]	Zn Content in Dose [g]	Impurities [g]	Residue after Evaporation [g]	Efficiency [%]	Ar_(g)_ Consumption [L]
**700**	20	3.03	2.94	0.09	0.09	**100**	**1.8**
	10	3.05	2.96	0.09	0.33	**91.9**	**0.9**
**800**	20	3.09	2.99	0.1	0.13	**99**	**1.8**
	10	3.08	2.98	0.1	0.13	**99**	**0.9**
	5	3.01	2.92	0.09	0.14	**98.3**	**0.45**

## Data Availability

Not applicable.

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
