# Peer review of "Theoretical and Practical Evaluation of the Feasibility of Zinc Evaporation from the Bottom Zinc Dross as a Valuable Secondary Material"

_materials, 2022, doi:10.3390/ma15248843_

Round 1
Reviewer 1 Report
The article presents an approach to the process of evaporating zinc as a valuable by-product. The topic is current and interesting from the scientific and research point of view. The article is methodological correctly written, the research analysis is clear, it is necessary to make a few clarifications and corrections as follows:
I recommend the authors to make some major changes in abstract:
The abstract is a little bit confuse and missis some information like more results and conclusions, I suggest to authors follow these rules:
- One or two sentences on BACKGROUND
- Two or three sentences on METHODS
- Less than two sentences on RESULTS
- One sentence on CONCLUSION
The Introduction section quite briefly refers to the content of the article, of course the authors pay attention to the key theses from the area of literature analysis, but this section should be more developed in terms of citation of sources. E.g. line 25, the information is detailed probably from a specific source, enter for citation.
The footnotes used are grouped, I encourage the authors to analyze the literature in more depth, not limiting itself to mentioning what individual authors presented in their works, it is advisable to refer to citations more thoroughly in a scientific way.
In lines 88-89 the author indicates that he received "saw dust" in the drilling process. Please elaborate on what exactly the process is. The statement that "saw dust" was obtained by means of a drill is non-technical............... in the drilling process we get as removed material - chips.
What was the chip fraction, does it matter for the process?
The description of the experiment procedure is presented in a concise scientific way, Figure 4 corresponds very poorly with the content, the scheme is very infantile for a scientific work, I suggest improving it or completely removing it.
Consistently, the author in the work refers in the text to what the drawings represent, doing so without in-depth analysis. E.g. line 185, everyone can read the caption under the drawing, the reference in the text should include an analysis of the drawing from the scientific and cognitive point of view. This absolutely needs to be improved throughout the work.
Figure 7 should be enlarged and analyzed in the text. In general, the Results and disscussion section is quite extensive, in some aspects very concise, e.g. Fig.7, reflects the sense of the research carried out in the correct way.
Conclusions do not refer to research results, they present what is in the article, lines 269-274 are surprisingly honest - they are not, however, conclusions of a scientific work.
I recommend writing the conclusions once again, in relation to the results obtained in the work - in a scientific way without literary descriptions.
Author Response
Dear Mr. or Mrs., please see the attachment.

Reviewer 2 Report
This manuscript described the theoretical and practical evaluation of zinc evaporation from bottom zinc dross (hard zinc) as a secondary zinc source. With the investigation of the calculation of HSC software and several experiments, the efficiency of zinc evaporation achieved almost 100%. This work has application value. But some analysis and expression need to be supplemented and improved, the authors should revise carefully.
1. In Figure 1, XRD figure cannot be seen clearly and authors should use the high-resolution image.
2. In Figure 2, what is the composition of particles with different shapes in the microstructure of bottom zinc dross, and whether they affect the evaporation process.
3. When analyzing Fe-Zn intermetallic compound, the reference cited by authors mentioned “ζ- FeZn13”, “δ FeZn10 phase”, “γ Fe-Zn”, “liquid Zn and αFe”, authors should add the phase analysis of intermediate products during evaporation and residue after evaporation to verify the accuracy of this conclusion.
4. The surface morphology of spectrum 1 is obviously different from that of spectrum 2 and spectrum 3. Figure 11 only shows the EDX results of spectrum 1. Is the EDX results of spectrum 2, spectrum 3 and spectrum 4 the same as that of spectrum 1?
5. Authors should explain the reason of the highest evaporation efficiency of Zn at 700℃ for 20 min.
6. There are some spelling or grammar errors in the manuscript, please check carefully. For example, “FeZn10 phase” in line 195.
7. What are the biggest advantages and innovations of this process compared with the existing zinc recovery techniques?
Author Response

(The authors gave the same response as above.)

Reviewer 3 Report
Theoretical and Practical Evaluation of the Feasibility of Zinc Evaporation from the Bottom Zinc Dross as a Valuable Secondary Material
The authors present an interesting work on recovering zinc from waste from galvanizing processes. They used thermodynamic processes (evaporation) and obtained almost pure materials (100% purity). Also performed the theoretical thermodynamic study. However, before the article is accepted, some points can be improved:
1) Discussion of results, based on existing literature. Some results are presented, without comparison with other papers on literature.
2) “The present study focusses on the development of the effective and economical way”. Is the process economically viable? Because the reuse of materials is always important, it is necessary to discuss this as well, as stated by the authors.
3) Conclusion is mixed with discussion. It needs to be rewritten, based on the objectives of the work
Other minor points are:
Line 87: Provide the field of activity of Slovak companies and the location (city).
Line 87-88: What is the granulometry collected?
Line 88 - 89: What drill size and the material? Provide the size of the collected grain (Zn sawdust).
Lines 90, 91, 92, 93: Provide information on the equipment used (brand and model).
Line 93, 100, 155: Captions should be self-explanatory. Improve the description.
Line 94, 114, 115, 220: In English, the decimal separator is a period (.), not a comma (,). Please check if they are correct in all Tables
Line 97: “Figure 1 XRD analysis of bottom zinc dross sample.” Include the database consulted to obtain the “REF. CODE” (reference).
Line 101: In “2.2 Analysis of Zinc Evaporating Conditions”. Some parts of this section should be moved to “Results”. By changing this section to Results, the authors must include other works in the Discussion.
Line 102: Please check if “HSC Chemistry program” is “HSC Chemistry software”.
Line 114: Table 2. The units could appear in the same line of the parameter. E.g.: Temperature (ºC); Equilibrium partial Pressure (Pa); Volume of Ar at 25 °C (dm³) etc.
Lines 133-134: What's the amount of argon in the system that allows the evaporation of zinc?
Line 144: “…when considering pure zinc in the system”. Please, correlate this result with that found in the XRD and EDX.
Line 151: Please consider changing “pictured” to “schematized”.
Line 167: Samples were collected in triplicates, quintuplicates? Please include this information.
Line 172: “Initial experiments were conducted under the temperature of 700°C”. Consider explaining in the methods section.
Author Response

(The authors gave the same response as above.)

Round 2
Reviewer 3 Report
The manuscript has been suitably improved, and this version, in my opinion, meets the parameters for publication in the journal Materials.
I congratulate the authors for the effort and work done.